# Analysis of 11,430 recombinant protein production experiments reveals that protein yield is tunable by synonymous codon changes of translation initiation sites

**Bikash K. Bhandari**[1]ʘ, **Chun Shen Lim**[1]ʘ, **Daniela M. Remus**[2], **Augustine Chen**[1], **Craig van Dolleweerd**[3], **Paul P. Gardner**[1,3]*

**1** Department of Biochemistry, School of Biomedical Sciences, University of Otago, Dunedin, New Zealand, **2** Callaghan Innovation Protein Science and Engineering, University of Canterbury, Christchurch, New Zealand, **3** Biomolecular Interaction Center, University of Canterbury, Christchurch, New Zealand

ʘ These authors contributed equally to this work.
* paul.gardner@otago.ac.nz

**Data Availability Statement:** Our code and data can be found in our GitHub repository (https://

## Abstract

Recombinant protein production is a key process in generating proteins of interest in the pharmaceutical industry and biomedical research. However, about 50% of recombinant proteins fail to be expressed in a variety of host cells. Here we show that the accessibility of translation initiation sites modelled using the mRNA base-unpairing across the Boltzmann's ensemble significantly outperforms alternative features. This approach accurately predicts the successes or failures of expression experiments, which utilised *Escherichia coli* cells to express 11,430 recombinant proteins from over 189 diverse species. On this basis, we develop TIsigner that uses simulated annealing to modify up to the first nine codons of mRNAs with synonymous substitutions. We show that accessibility captures the key propensity beyond the target region (initiation sites in this case), as a modest number of synonymous changes is sufficient to tune the recombinant protein expression levels. We build a stochastic simulation model and show that higher accessibility leads to higher protein production and slower cell growth, supporting the idea of protein cost, where cell growth is constrained by protein circuits during overexpression.

## Author summary

Recombinant proteins are widely used as therapeutics, such as vaccines, monoclonal antibodies, hormones and enzymes. However, the success rate of recombinant protein production is about 50%. To address this problem, we propose optimising the unpairing propensities of nucleotides around translation initiation sites using a thermodynamic quantity called mRNA accessibility. Our study shows that this method is generalisable across prokaryotic and eukaryotic expression hosts. Importantly, we validated this method using laboratory experiments and computational modelling. Furthermore, we propose a

github.com/Gardner-BinfLab/TIsigner_paper_2019). These include the scripts and Jupyter notebooks to reproduce our results and figures. The source code of TIsigner is available at https://github.com/Gardner-BinfLab/TISIGNER-ReactJS. The public web version of this tool runs at https://tisigner.com/tisigner. The experimental data, analysis and results are available at https://github.com/bkb3/TIsignerExperiment/tree/master/Jupyter and an interactive version of results are available at https://bkb3.github.io/TIsignerExperiment/.

**Funding:** This work was supported in part by the Ministry of Business, Innovation and Employment (https://www.mbie.govt.nz/) [MBIE Smart Idea grant: UOOX1709 to P.P.G. and C.D., and MBIE Data Science Programmes grant: UOAX1932 to P.P.G.] and the Royal Society of New Zealand Te Apārangi (https://www.royalsociety.org.nz/) [Marsden grant: 19-UOO-040 to P.P.G.]. B.K.B was also supported in part by the University of Otago Postgraduate Publishing Bursary (Doctoral). The funders had no role in study design, data collection and analysis, decision to publish, or preparation of the manuscript.

**Competing interests:** The authors have declared that no competing interests exist.

low cost technique to tune protein expression by engineering minimal changes to genes of interest through our web application (https://tisigner.com/tisigner).

## Introduction

Recombinant protein expression has numerous applications in biotechnology and biomedical research. Despite extensive refinements in protocols over the past three decades, half of the experiments fail in the expression phase (http://targetdb.rcsb.org/metrics/). Notable problems are the low expression of 'difficult-to-express' proteins such as those found in, or associated with, membranes, and the poor growth of the expression hosts, which may relate to toxicity of heterologous proteins [1] (see [2, 3] for detailed reviews). Despite these issues, mRNA abundance only explains up to 40% of the variation in protein abundance, presumably due to variation in translation and turnover rates [4–10].

For *Escherichia coli*, mainstream models that may explain the lower-than-expected correlation between mRNA and protein levels are codon-usage and mRNA structure. Codon analysis is based on the frequency of codon usage in highly expressed proteins using codon adaptation index (CAI) [11] or tRNA adaptation index (tAI)—these are thought to capture tRNA availability which may influence translation rates [12, 13]. On the other hand, stable mRNA structures are thought to impede the assembly and progress of ribosomes on mRNAs [14–16]. More recent studies show stronger support for models based on mRNA folding, in which the stability of RNA structures, usually estimated using nearest-neighbour minimum-free energy (MFE) models, around the Shine-Dalgarno sequence and translation initiation site (e.g., AUG start codon) inversely correlates with protein expression [15–20]. We recently proposed a third model in which the avoidance of inappropriate interactions between mRNAs and non-coding RNAs (ncRNAs) has a strong effect on protein expression [21]; in addition, we evaluate a related measure 'accessibility' which considers all possible intramolecular base-unpairing probabilities [22]. Many of these features are interdependent, which presents a major challenge for identifying useful features.

The existing algorithms for gene optimisation that sample synonymous protein-coding sequences use models based on CAI, tAI, MFE, and/or G+C content (%) [23–27]. However, these models are usually evaluated on relatively small numbers of endogenous proteins, reporter proteins, or heterologous proteins with synonymous variants, often with poor separation of training and test datasets. It is unclear whether these features are generalisable to explain the expression of all heterologous proteins. To address this question, we have used data from multiple large-scale, non-redundant recombinant protein production experiments (N = 11,430), proteomics (N = 3,725), and fluorescent reporters (N = 82,002) from bacterial and eukaryotic species in order to identify mRNA features that best explain variation in protein abundance. This problem has not previously been investigated with this scale of heterogeneous datasets in any previous study.

We find that mRNA accessibility is a single best predictor of protein expression across the datasets, and accurately predicts the successes and failures of 11,430 experiments of recombinant protein expression in *E. coli*. Specifically, the accessibility of translation initiation sites outperform other mRNA features by capturing all possible optimal or suboptimal structures beyond translation initiation sites. With this information, we propose how accessibility can be exploited to fine-tune recombinant protein expression at a low cost. Specifically, we built a web server called TIsigner (Translation Initiation coding region designer), which optimises a protein-coding sequence by suggesting synonymous codon changes within the first nine

codons. Therefore, our approach makes gene optimisation accessible, as PCR can be used rather than an expensive full-length gene synthesis.

## Results

### Accessibility of translation initiation sites strongly correlates with protein abundance

To identify an accurate model of mRNA structure that explains protein expression, we examined an *E. coli* expression dataset of green fluorescent protein (GFP) fused in-frame with a library of 96-nt upstream sequences (N = 244,000 variants) [16]. These 96-nt sequences were generated to achieve a full factorial design by varying A+T content (%), CAI, codon ramp bottleneck position and strength, hydrophobicity of the encoded peptide, and minimum free energy (MFE). We removed redundancy of these 96-nt upstream sequences by clustering on sequence similarity, giving rise to 14,425 representative sequences. We calculated the accessibility (also known as 'opening energy' based on unpairing probability) for all the corresponding sub-sequences (Fig 1, see the definitions and equations in Methods).

Previous studies have defined mRNA accessibility using different terms [14, 24, 28–33] (see Additional notes, S2 File). Here we use a partition function method implemented in RNAplfold to calculate accessibility [22] (Fig 1, see Methods). We examined the correlation between the opening energies and GFP levels. We found that the opening energies of translation initiation sites, in particular from the nucleotide positions −30 to 18 (−30:18), shows the highest correlation with protein abundances (Fig 2A; Spearman's correlation, $R_s$ = −0.65, P < $2.2 \times 10^{-16}$). This is stronger than the highest correlation between the MFE at the region −30:30 (MFE −30:30) and protein abundance, which was previously reported as the highest

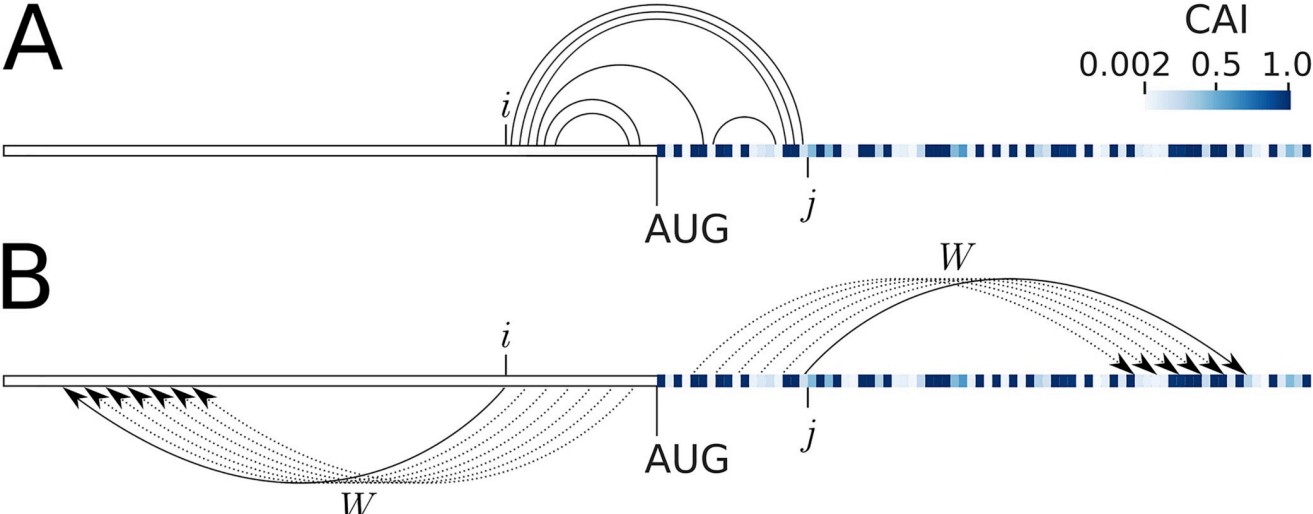

**Fig 1. Opening energy (accessibility) has greater contextual information than minimum free energy (MFE).** Schematic representation and interpretation of MFE and opening energy of a mRNA sequence (GFP). Codons are color coded using the weights of Codon Adaptation Index (CAI) from Sharp and Li (1987) [11]. A: The computation of MFE of the region $i \ldots j$ results in finding a single most stable structure. This stable structure contains pairings within the region as indicated by arcs. Hence, this approach is unable to detect any information beyond the target region (initiation site in this case). For example, a change in CAI after the nucleotide at position $j$ does not affect the MFE. In addition, the predicted single structure may not be present under physiological conditions. B: The computation of opening energy of the region $i \ldots j$ uses several windows, each of length $W$ nucleotides, shown by dotted and solid arrows (flanking window). The partition function used to determine the opening energy is computed over all possible structures (optimal and suboptimal) from the Boltzmann's ensemble where the region $i \ldots j$ is unpaired (Methods). As these windows could be extended well beyond the target region, opening energy contains additional contextual information. For example, a change of CAI beyond the nucleotide $j$ could influence the opening energy (Results, Accessibility captures the full ensemble average energy of a sequence).

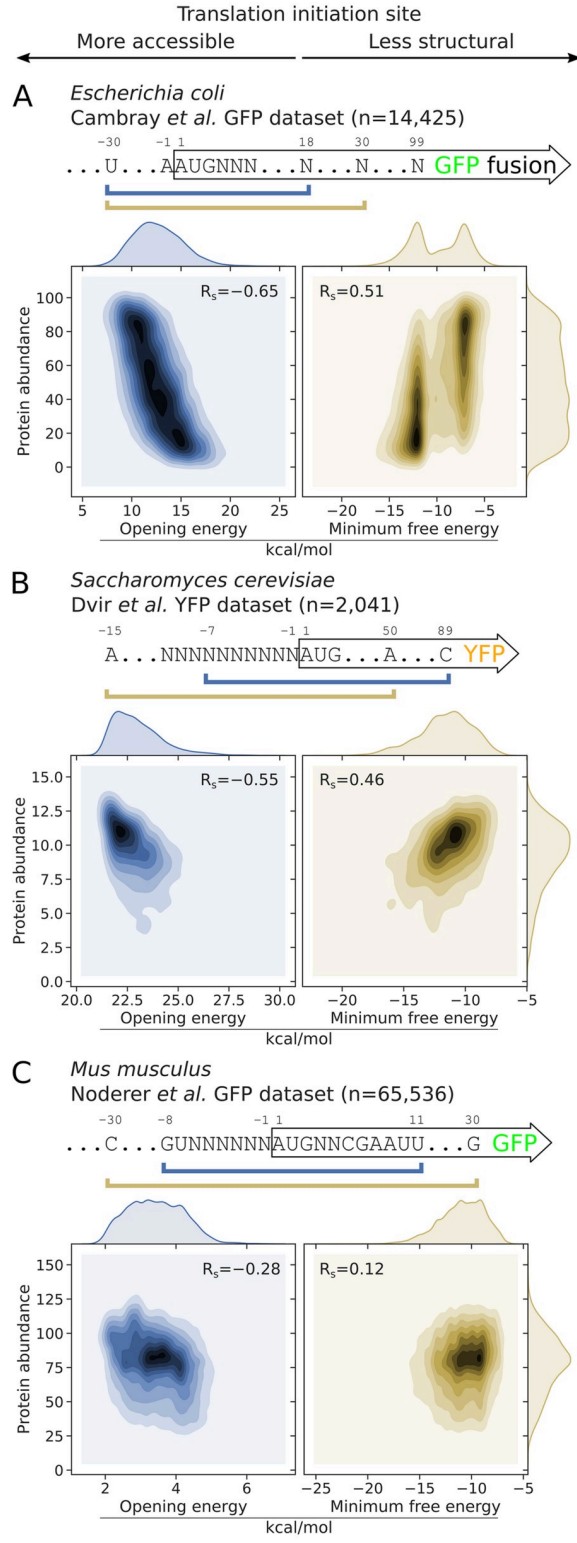

**Fig 2. Correlations between the opening energies of translation initiation sites and protein abundances are stronger than that of minimum free energies (MFE).** A: For *E. coli*, the opening energy at the region −30:18 shows the strongest correlation with protein abundance (see also Fig 3A and 3B, or Fig A in S1 Fig, sub-sequence l = 48 at position i = 18). For this analysis, we used a representative green fluorescent protein (GFP) expression dataset from Cambray et al. (2018) [16]. The reporter library consists of GFP fused in-frame with a library of 96-nt upstream sequences (N = 14,425). The MFE at the region −30:30 (MFE −30:30) shown was determined by Cambray et al. (right panel). B:

For *S. cerevisiae*, the opening energy −7:89 shows the strongest correlation with protein abundance (see also Fig B in S1 Fig, sub-sequence l = 96 at position i = 89. For this analysis, we used the yellow fluorescent protein (YFP) expression dataset from Dvir et al. (2013) [19]. The YFP reporter library consists of 2,041 random decameric nucleotides inserted at the upstream of YFP start codon. The MFE −15:50 was previously shown to correlate the best with protein abundance (right panel). C: For *M. musculus*, the opening energy −8:11 shows the strongest correlation with protein abundance (see also Fig C in S1 Fig, sub-sequence l = 19 at position i = 11). For this analysis, we used the GFP expression dataset from Noderer et al. (2014) [34]. The GFP reporter library consists of 65,536 random hexameric and dimeric nucleotides inserted at the upstream and downstream of GFP start codon, respectively. The MFE −30:30 was shown (right panel). See also S1 File. $R_s$, Spearman's rho. The Bonferroni adjusted P-values are below the machine's underflow level for the correlations between opening energies and protein abundances shown in the left panels.

ranked feature [Fig 2A; $R_s$ = 0.51, P < 2.2 × 10$^{-16}$ (right panel)]. To account for multiple-testing, the P-values were adjusted using Bonferroni's correction and reported to machine precision.

We repeated the analysis for a dataset of yellow fluorescent protein (YFP) expression in *Saccharomyces cerevisiae* [19]. This dataset corresponds to a library of 5′UTR variants, in which the 10-nt sequences preceding the YFP translation initiation site were randomly substituted (N = 2,041 variants). In this case, the opening energy −7:89 showed a stronger correlation with protein abundance than that of the MFE −15:50 reported previously (Fig 2B; $R_s$ = −0.55 versus 0.46).

To examine the usefulness of accessibility in complex eukaryotes, we analysed a dataset of GFP expression in *Mus musculus* [34]. The reporter library was originally designed to measure the strength of translation initiation sequence context, in which all possible substitutions were made at the flanking regions of the GFP translation initiation site (6-nt upstream region and 2-nt downstream region of initiation codon; N = 65,536 variants). Here the opening energy −8:11 showed a maximum correlation with expressed proteins, which again, is stronger than that of the MFE −30:30 (Fig 2C; $R_s$ = −0.28 versus 0.12).

Taken together, our findings suggest that the accessibility of translation initiation sites strongly correlates with protein abundance across species. Interestingly, our findings in *E. coli* also suggest that the surrounding region of initiation sites, including the Shine-Dalgarno sequence [35] at −13:−8, should be accessible, presumably in order to recruit ribosomes. In contrast, the Shine-Dalgarno sequence is absent in yeasts and complex eukaryotes, which may explain why the computed accessibility regions begin at positions $\geq$ −8. In eukaryotes, the 43S preinitiation complexes scan from the 5′-cap end of the mRNAs [36]. This mechanism employs helicases such that the RNA structures preceding initiation codons are scanned through. However, caution is in order here, as large-scale recombinant protein production datasets for these eukaryotes are not available to validate these findings. Further investigation into the differences in the mechanisms of translation initiation between prokaryotes and eukaryotes would be useful to explain why these mRNA regions are distinct.

Theoretically, bacterial 30S subunits can initiate at any position as non-AUG initiation codons are more common in bacteria than eukaryotes, and most bacterial mRNAs are polycistronic. However, in agreement with previous high-throughput RNA structural probing studies, we found that the regions −30:30 of bacterial mRNAs are significantly less structured and are A-rich [37–40]. We reasoned that accessibility is likely more important in bacteria than eukaryotes (Fig 2, stronger correlation between accessibility and protein abundance). High accessibility of initiation sites likely improves a greater selectivity in translation initiation in bacteria.

## Accessibility predicts the outcome of recombinant protein expression

We investigated how accessibility performs in the real world in prediction of recombinant protein expression. For this purpose, we carefully curated and analysed 11,430 expression

experiments in *E. coli* from the 'Protein Structure Initiative:Biology' (PSI:Biology) [41–44]. These PSI:Biology targets were expressed using the pET21_NESG expression vector that harbours the T7lac inducible promoter and a C-terminal His tag [43].

We divided the experimental results of the PSI:Biology targets into protein expression 'success' and 'failure' groups that were previously curated by DNASU (8,780 'Protein_Confirmed' and 2,650 'Tested_Not_Found' determined by SDS-PAGE analysis, respectively; see S2 Fig). These PSI:Biology targets span more than 189 species and the failures are representative of various problems in heterologous protein expression. Only 1.6% of the targets were *E. coli* proteins, which is negligible (N = 179; see S2 Fig).

We calculated the opening energies for all possible sub-sequences of the PSI:Biology targets as above (Fig 3, positions relative to initiation codons). For each sub-sequence region, we used the opening energies to predict the expression outcomes and computed the prediction accuracy using the area under the receiver operating characteristic curve (AUC; see Fig 3C). A closer look into the correlations between opening energies and expression outcomes, and AUC scores calculated for the sub-sequence regions reveals a strong accessibility signal of translation initiation sites (Fig 3B and 3C, Cambray's GFP and PSI:Biology datasets, respectively). We matched the correlations and AUC scores by sub-sequence regions and confirmed that sub-sequence regions that have strong correlations are likely to have high AUC scores (Fig 3D). In contrast, the sub-sequence regions that have zero correlations are not useful for predicting the expression outcomes (AUC approximately 0.5).

We then asked how accessibility manifests in the endogenous mRNAs of *E. coli*, for which we studied a proteomics dataset of 3,725 proteins available from PaxDb [45]. As expected, we observed a similar accessibility signal, with the region −25:16 correlated the most with protein abundance (Fig 3E, sub-sequence l = 41 at position i = 16). However, the correlation was rather low (R = −0.17, P < 2.2 × 10^{−16}), which may reflect the limitation of mass spectrometry to detect lower abundances [46, 47]. Furthermore, the endogenous promoters have variable strength, which gives rise to a broad range of mRNA and protein levels [48, 49]. Taken together, our results show that the accessibility signal of translation initiation sites is very consistent across various datasets analysed (Fig 3 and S1 Fig).

## Accessibility outperforms other features in prediction of recombinant protein expression

To choose an accessibility region for subsequent analyses, we selected the top 200 regions from the above correlation analysis on Cambray's GFP dataset (Fig 3B) and used random forest to rank their Gini importance scores in prediction of the outcomes of the PSI:Biology targets. The region −24:24 was ranked first (Fig 3B, l = 48 and i = 24), which is nearly identical to the region −23:24 with the top AUC score (Fig 3C, l = 47 and i = 24, AUC = 0.70). We therefore used the opening energy at the region −24:24 in subsequent analyses (Fig 4A). Interestingly, both the Shine-Dalgarno sequence (−13:−8 or l = 21 and i = −8) and initiation codons (l = 3 and i = 3) have weaker correlations and AUC scores than the region −24:24 (Fig 3B, 3C and 3E). This suggests that a slightly larger region around these key motifs provides a better context of accessibility and thus a better prediction of protein expression.

We asked how the other features perform compared to accessibility in prediction of heterologous protein expression, for which we analysed the same PSI:Biology dataset. We first calculated the MFE and avoidance at the regions −30:30 and 1:30, respectively. These are the local features associated with translation initiation rate. We also calculated CAI [11], tAI [50], codon context (CC) [51], G+C content, and Iχnos scores [52]. CC is similar to CAI except it takes codon-pairs into account, whereas the Iχnos scores are translation elongation rates

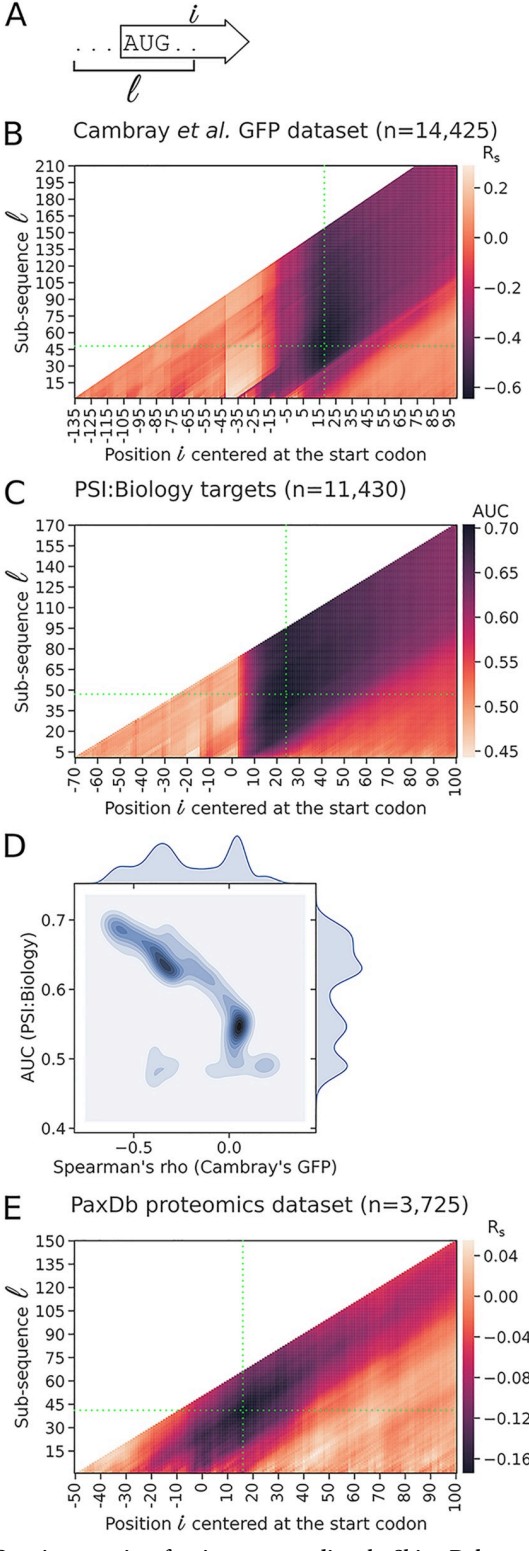

**Fig 3. Opening energies of regions surrounding the Shine-Dalgarno and start codons are predictive of protein expression in *E. coli*.** A: Schematic representation of a transcript sub-sequence l at position i for the calculation of opening energy. For example, the sub-sequence l = 10 at position i = 10 corresponds to the region 1:10. B: Correlations between the opening energies for the sub-sequences of GFP transcripts and protein abundances. The opening energy at the region −30 to 18 nt (sub-sequence l = 48 at position i = 18, green crosshair) shows the strongest correlation with

protein abundance [$R_s$ = −0.65; N = 14,425, GFP expression dataset of Cambray et al. (2018)]. For this dataset, the reporter plasmid used is pGC4750, in which the promoter and ribosomal binding site are oFAB1806 inducible promoter and oFAB1173/BCD7, respectively. C: Prediction accuracy of the expression outcomes of the PSI:Biology targets using opening energy (N = 11,430). The opening energy at the region −23:24 (sub-sequence l = 47 at position i = 24, green crosshair) shows the highest prediction accuracy score (AUC = 0.70). For this dataset, the expression vector used is pET21_NESG, in which the promoter and fusion tag are T7lac and C-terminal His tag, respectively. D: Comparison between the correlations and AUC scores by sub-sequence region taken from the above analyses. The sub-sequence regions that have strong correlations are likely to have high AUC scores, whereas the sub-sequence regions that have no correlations are likely not useful in prediction of the expression outcomes. E: Correlations between the opening energies for the sub-sequences of *E. coli* transcripts and protein abundances. The transcripts used for this analysis are protein-coding sequences concatenated with 50 and 10 nt located upstream and downstream, respectively. The opening energy at the region −25:16 (sub-sequence l = 41 at position i = 16, green crosshair) shows the strongest correlation with protein abundance ($R_s$ = −0.17; N = 3,725, PaxDb integrated proteomics dataset). See also S1 File. $R_s$, Spearman's rho.

predicted using a neural network model trained with ribosome profiling data (S3 Fig). These are the global features associated with translation elongation rate. We built a random forest model to rank the Gini importance scores of these local and global features. The local features ranked higher than the global features (Fig 4B). We then calculated and compared the prediction accuracy of these features. The AUC scores for the local features were 0.70, 0.67 and 0.62 for the opening energy, MFE and avoidance, respectively, whereas the global features were 0.58, 0.57, 0.54, 0.54 and 0.51 for Iχnos, G+C content, CAI, CC and tAI, respectively (Fig 4C). The local features outperform the global features, suggesting that effects on translation initiation are a major predictor of the outcome of heterologous protein expression. We further examined the local G+C contents corresponding to the local features (S4 Fig). The G+C contents in the regions −24:24 and −30:30 weakly correlate with opening energy and MFE, respectively. The AUC scores for these local G+C contents are also lower than the corresponding local features, suggesting that these local G+C contents are not good proxies for the corresponding local features. Overall, our findings support previous reports that the effects on translation initiation are rate-limiting [15, 20] which, interestingly, correlate with the binary outcome of recombinant protein expression (Fig 4D). Importantly, accessibility significantly outperformed all other features (Fig 4C, see confidence intervals of AUC scores).

To identify a good opening energy threshold, we calculated positive likelihood ratios for different opening energy thresholds using the cumulative frequencies of true negative, false negative, true positive and false positive derived from the above receiver operating characteristic (ROC) analysis (top panel in S5 Fig). Meanwhile, we calculated the 95% confidence intervals of these positive likelihood ratios using 10,000 bootstrap replicates. We reasoned that there is an upper and lower bound on translation initiation rate, therefore the relationship between translation initiation rate and accessibility is likely to follow a sigmoidal pattern. We fit the positive likelihood ratios into a four-parametric logistic regression model (S5 Fig). As a result, we are 95% confident that an opening energy of 10 kcal/mol or below at the region −24:24 is about two times more likely to belong to the sequences which are successfully expressed than those that failed. To allow easy interpretation of results, we derived 'Expression Score' from this logistic regression curve with a scale from (see Methods).

## Accessibility captures the full ensemble average energy of a sequence

To illustrate the advantage of accessibility over MFE, we analysed a proteomic dataset of *E. coli* cells infected with bacteriophage T7 [53]. The major capsid gene was codon-deoptimised to generate a mutant T7 strain [53]. Specifically, the first and the last 14 codons of the major

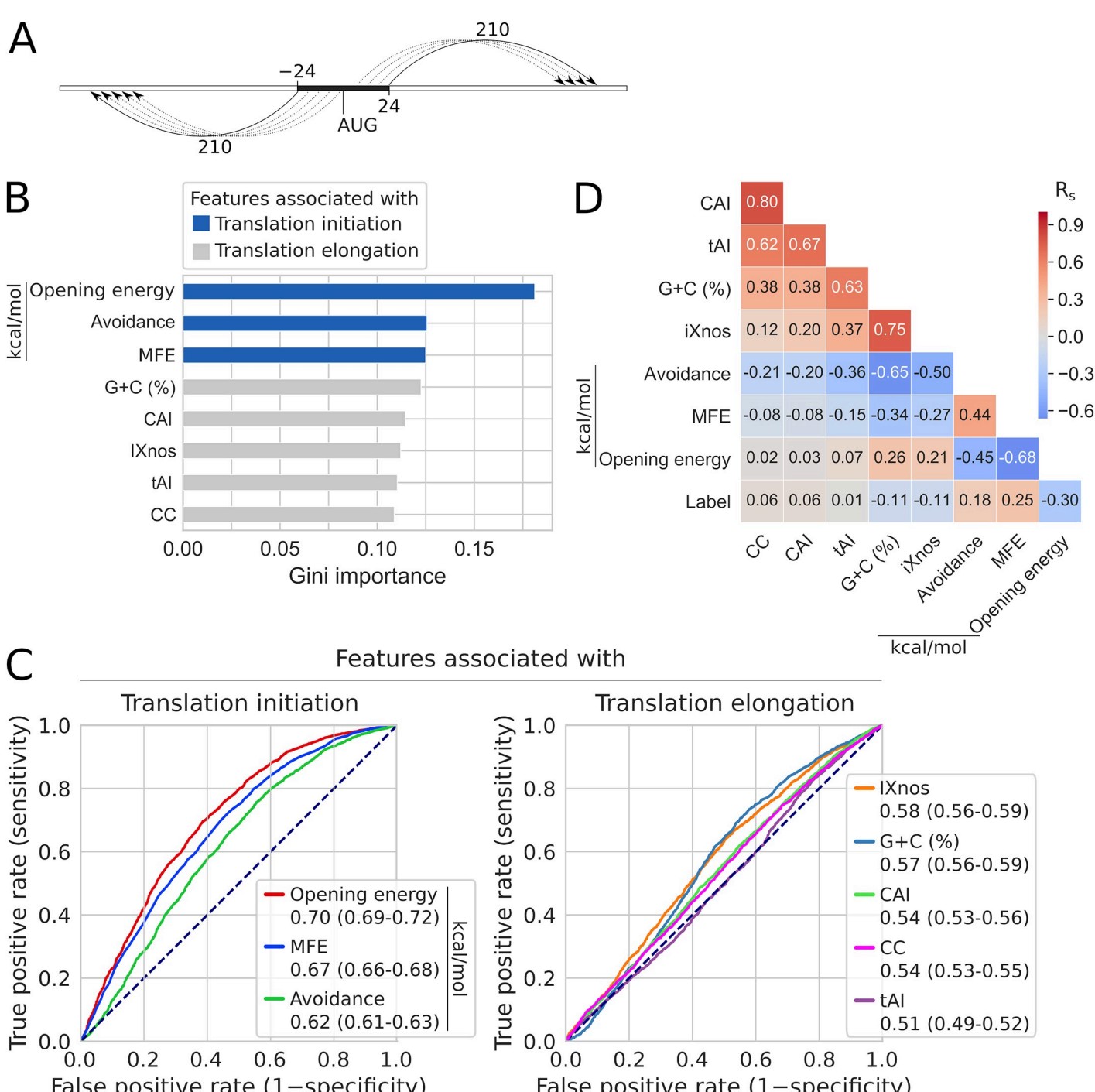

**Fig 4. Accessibility of translation initiation sites is the strongest predictor of heterologous protein expression in *E. coli*.** A: A partition function approach to compute the opening energy of the region −24:24 (solid black) in this analysis. Each window (arrow) is of the length of 210 nucleotides. The solid arrows represent the flanking windows. The dotted arrows represent other windows in between. Thus, the computation of opening energy for the region −24:24 captures the unpairing propensities of the surrounding region. It should be noted that the sliding window is constrained by the lengths of the flanking sequences. This partition function approach can be customised and executed using the algorithm implemented in RNAplfold. B: mRNA features ranked by Gini importance for random forest classification of the expression outcomes of the PSI:Biology targets (N = 8,780 and 2,650, 'success' and 'failure' groups, respectively). The features associated with translation initiation rate (blue; opening energy −24:24, minimum free energy (MFE) −30:30, and mRNA:ncRNA avoidance 1:30) have higher scores than the feature associated with translation elongation rate [grey; tRNA adaptation index (tAI), codon context (CC), codon adaptation index (CAI), G+C content (%), and Iχnos]. The Iχnos scores are translation elongation rates predicted using a neural network model trained with ribosome profiling data (S3 Fig). C: ROC analysis shows that accessibility (opening energy −24:24) has the highest classification accuracy. The AUC scores with 95% confidence intervals are shown. See also S1 File. D: Accessibility (opening energy −24:24) is the best feature in explaining the expression outcomes. Outcomes are represented with 'Label'. For this dataset, these labels were binary. MFE, Minimum Free Energy; $R_s$, Spearman's rho.

capsid gene remained unchanged, making this a good case study to compare accessibility, MFE, and CAI.

We calculated the opening energies, MFEs and CAIs of the wild-type and mutant sequences (S6 Fig, orange and blue colours, respectively). We also scored the wild-type and mutant sequences using the above logistic regression curve, and obtained the approximated 'Expression Scores' of 89 and 38, respectively (opening energies of 9.05 kcal/mol and 13.76 kcal/mol, respectively). The MFEs of the wild-type and mutant sequences are the same because the local regions −30:30 are identical.

In contrast to MFE, accessibility was able to make an accurate prediction by capturing the full ensemble average energy of the mutant sequence. Importantly, although we use the region −24:24 for the accessibility computation, we are able to capture the key propensity beyond this region. This is because the computation of the partition function, thus the opening energy, for the region −24:24, also utilises the surrounding region (here 210 nucleotides around −24:24, see Methods and Fig 4A). This unique approach makes opening energy more robust than the traditional MFE approach. These results also highlight that mRNA features are interrelated (accessibility and CAI in this case), as such a careful factorial design is necessary to identify the causal features [16, 54].

## Accessibility can be improved using a simulated annealing algorithm

The above results suggest that accessibility can, in part, explain the low expression problem of heterologous protein expression. Therefore, we sought to exploit this idea for optimising gene expression. Due to the lack of open source libraries/packages specialised for sequence optimisation, we developed a simulated annealing (Metropolis-Hastings) based algorithm to maximise the accessibility at the region −24:24 using synonymous codon substitution (see Code and data availability, our custom JavaScript and Python modules for the web server and command line tool, respectively). Previous studies have found that full-length synonymous codon-substituted transgenes may produce unexpected results, such as a reduction in mRNA abundance, RNA toxicity, and/or protein misfolding [21, 52, 55, 56]. Therefore, we sought to determine the minimum number of codons required for synonymous substitutions in order to achieve near-optimum accessibility. For this purpose, we used the PSI:Biology targets that failed to be expressed. We applied our simulated annealing algorithm such that synonymous substitutions can happen at any codon of the sequences except the start and stop codons (S7 Fig), although the changes may not necessarily happen to all codons due to the stochastic nature of our optimisation algorithm (see Methods). Next, we constrained synonymous codon substitution to the first 14 codons and applied the same procedure (Fig A in S7 Fig). Therefore, the changes may only occur at any or all of the first 14 codons. We repeated the same procedure for the first nine and also the first four codons. Thus a total of four series of codon-substituted sequences were generated. We then compared the distributions of opening energy −24:24 for these series using the Kolmogorov-Smirnov statistic ($D_{KS}$; see Fig B in S7 Fig). The distance between the distributions of the nine and full-length codon-substituted series was significantly different yet sufficiently close ($D_{KS} = 0.087$, $P = 3.3 \times 10^{-8}$), suggesting that optimisation of the first nine codons is sufficient in most cases to achieve an optimum accessibility of translation initiation sites. We named our software Translation Initiation coding region designer (TIsigner), which by default, allows synonymous substitutions in the first nine codons.

We asked to what extent the existing gene optimisation tools modify the accessibility of translation initiation sites. For this purpose, we first submitted the PSI:Biology targets that failed to be expressed to the ExpOptimizer web server from NovoPro Bioscience (see

Methods). We also optimised the PSI:Biology targets using the standalone version of Codon Optimisation OnLine (COOL) [26]. We found that both tools increase accessibility indirectly even though their algorithms are not specifically designed to do so. In fact, a purely random synonymous codon substitution on these PSI:Biology targets using our own script resulted in similar increases in accessibility (Fig C in S7 Fig). These results may explain some indirect benefits from the existing gene optimisation tools (i.e. any change from suboptimal is likely to be an improvement, see below).

## Low protein yields can be improved by synonymous codon changes in the vicinity of translation initiation sites

To demonstrate that heterologous protein expression is tunable with minimum effort, we designed and tested a series of GFP reporter gene constructs. We tested 29 plasmids harbouring GFP reporter genes with synonymous changes within the first nine codons (opening energies of 5.56–21.68 kcal/mol; Tables A-C in S2 File; and S3 File). GFP expression is controlled by an IPTG (isopropyl-β-D thiogalactopyranoside) inducible T7lac promoter. In addition, all plasmids harbour a second reporter gene (mScarlet-I), which is controlled by the constitutive promoter from the nptII gene for aminoglycoside-3′-O-phosphotransferase of *E. coli* transposon Tn5 [57, 58]. mScarlet-I expression was measured to correct for plasmid copy number and as a proxy for bacterial growth [59].

Consistent with the above results, the GFP level significantly correlates with accessibility (i.e., anti-correlates with opening energy, $R_s = -0.53$, $P = 3.4 \times 10^{-3}$; Fig 5A). This correlation was also the strongest compared to other features, which independently supports our observations on multiple large-scale datasets. Curiously, we observed a diminishing return with opening energies lower than that of the wild-type sequence (11.68 kcal/mol). To investigate this, we simulated a protein production experiment by modelling cell growth, transcription, translation, and turnovers (see Methods). We assumed that opening energy of 12 kcal/mol or below is favourable in this model, based on our analysis of 8,780 PSI:Biology 'success' group (S7 Fig). Interestingly, this stochastic model shows a similar protein production trend as the actual experiment (Fig 5B). Surprisingly, this in silico model also shows that an efficient protein production leads to slower cell growth (Fig 5B). This phenomenon, also known as protein cost, is observed in vivo, in which overexpression slows down cell growth due to the cost-benefit trade-offs of protein circuits [60–66].

Additionally, we tested this finding using the luciferase reporter from *Renilla reniformis* (RLuc). We designed and tested a series of nine RLuc variants with higher accessibility than the wild-type sequence, in which sequence optimisations were performed within the first 9 codons (opening energies of 5.77–10.38 kcal/mol; S8 Fig and S3 File). We also tested five commercially designed sequences, which incorporated sequence optimisations across the entire RLuc gene. The TIsigner optimal sequence (5.77 kcal/mol) was expressed at a higher level compared to a TIsigner suboptimal sequence (10.38 kcal/mol) and the wild type (13.15 kcal/mol) in the BL21Star(DE3) *E. coli* host. However, it is worth noting that a high proportion of the expressed RLuc protein is insoluble, as analysed by SDS-PAGE (Fig A in S8 Fig). We also carried out luciferase reporter assay to compare the levels of active soluble RLuc but detected no significant differences between the luminescence levels of these three sequences. The poor correlation between RLuc protein abundance with luciferase activity could be partly attributed to the observed aggregation problems in Fig A in S8 Fig.

Nevertheless, a TIsigner suboptimal sequence (9.90 kcal/mol) and the commercially optimised sequences did produced approximately 1.5 times higher luminescence than the wild-type (Fig B in S8 Fig), an increase which is still insufficient for purification and recombinant

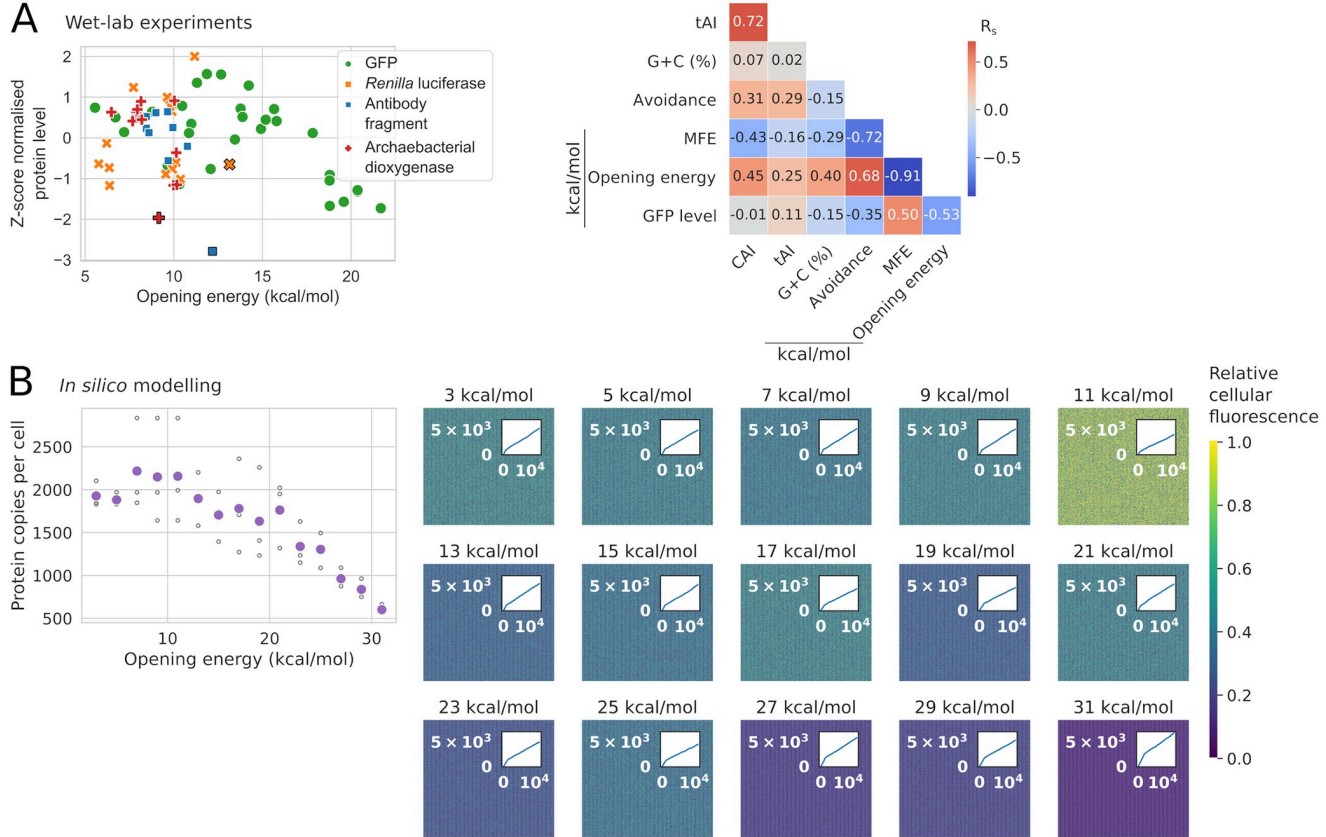

**Fig 5. The yields of heterologous protein productions are tunable by synonymous codon changes in the first nine codons.** A: GFP level strongly correlates with accessibility, i.e., anti-correlates with opening energy ($R_s = -0.53$, P = $3.4 \times 10^{-3}$; N = 29). This correlation is the strongest compared to other features (right), which independently supports the observations on multiple large-scale datasets (Figs 2–4). The protein levels of GFP, *Renilla* luciferase (RLuc), an antibody fragment and an archaebacterial dioxygenase were transformed using a z-score method. The GFP and RLuc levels were derived from the average values of at least two and three independent biological replicates, respectively. Black outlines denote wild-type sequences. See also S8 and S9 Figs and S3 File. CAI, codon adaptation index; CC, codon context; $R_s$, Spearman's rho; tAI, tRNA adaptation index. B: Stochastic simulation of a protein production experiment (e.g., GFP) by modelling cell growth, transcription, translation, and turnovers, given that translation initiation sites with opening energies less than or equal to 12 kcal/mol is optimum. The in silico model shows a similar trend of protein production as the wet-lab experimental results. Unfilled and filled (purple) circles denote the in silico replicates and their corresponding average values, respectively ($R_s = -0.75$, P = $2.8 \times 10^{-9}$). Similar to an actual recombinant production experiment, the in silico model also shows that efficient protein production (higher relative cellular fluorescence) leads to slower cell growth and vice versa (right, see insets for the opening energies of 11 kcal/mol versus 31 kcal/mol). Insets represent the number of cells with iterations.

protein production standard. It is worth noting that this TIsigner sequence (9.90 kcal/mol) harbours only two nucleotide changes compared to 187 to 241 nucleotide changes for the commercially designer sequences (0.2% versus 20.0%-25.7% of the full-length sequence). Due to the persisting aggregation issues, further testing in RLuc reporter for the full spectrum of opening energies is no longer warranted.

As both wild-type GFP and RLuc proteins were expressed at high levels in *E. coli*, we posited whether poorly expressed proteins can also be improved by increasing accessibility of translation initiation sites. We performed densitometric analysis of previously published Western blots using imageJ [67], which include the results of a cell-free expression system using constructs harbouring a wild-type antibody fragment or archaebacterial dioxygenase and its synonymous variants (within the first six codons) [30]. Indeed, we observed variants with opening energies lower than the wild-type sequences were expressed at higher levels (S9 Fig).

A recent study also showed that an increase in accessibility of a 30 bp region from the Shine-Dalgarno sequence enhances the expression level of human voltage dependent anion channel in an *E. coli* cell-free system, which further supports our findings [68]. Overall, the findings show that optimising accessibility is useful for tuning protein expression in both cellular and cell-free expression systems.

## Discussion

### Accessibility is a single sequence feature that explains most of the variation in protein abundance

Our data-driven approach shows that the accessibility of translation initiation sites is the strongest predictor of heterologous protein expression in *E. coli*. However, protein expression is inherently noisy due to the interplay of many cellular processes. At the transcript level, many mRNA features are not truly independent (Fig 4, e.g., accessibility, MFE, and G+C content), which aggravates the problem of identifying the key features. As such, a careful design of experiments such as using factorial methods for generating mRNA sequences is crucial for a complete traversal of the feature landscape. Due to the large-scale nature of such factorial designs, to-date few attempts have been made, e.g., 244,000 GFP and 86 firefly luciferase synonymous variants tested in *E. coli* and HeLa cells, respectively [16, 54]. These fluorescence reporter studies concluded that MFE was the best predictor but with modest correlations (e.g., Fig 2, Spearman's correlations of 0.51 in E. coli). These modest correlations reflect the noisiness of the system which further poses a problem for obtaining a better predictor. Furthermore, MFE estimation involves identifying the thermodynamically most probable structure from Boltzmann's ensemble, which is often inaccurate in a biological system where different constraints may prevent a mRNA from attaining the most probable conformation.

With this in mind, we used opening energy, an accessibility-based approach that takes the full ensemble average energy into account. This includes all possible RNA structures, including suboptimal structures that are not reported by MFE models by default [22, 69]. Indeed, our approach gave us a better correlation from multiple datasets where MFE was previously concluded to be a better predictor. We have shown that accessibility is superior to MFE even for the datasets without factorial designs such as the PSI:Biology dataset (Fig 4), where the feature space is sampled irregularly, and the expression levels of recombinant proteins were categorised into 'Tested_Not_Found' and 'Protein_Confirmed' with SDS-PAGE analysis [42, 44] (S2 Fig, 11,430 proteins from over 189 diverse species).

Moreover, the correlation between endogenous mRNA and protein levels is limited in both bacteria and eukaryotes (0.4–0.7) [10, 70–75], where theoretically mRNA levels should provide an upper-bound on correlation statistics of mRNA features. Besides mRNA level, accessibility is a sequence feature that explains most of the variation in protein abundance (Fig 2, $R_s$ of 0.28–0.65). Any further improvements in correlations are likely to be hindered by the noise and encountered diminishing returns.

### Adoption of accessibility for tuning protein expression

The accessibility of a region of mRNA can be understood as the ability of that region to base-pair with other nucleotides, including the flanking region. There are two distinct ways to define the accessibility of a region. (i) The first way is to consider the minimum Gibbs free energy (MFE) of the region. A lower value of MFE implies a stronger folding at that region. Hence, the region is less likely to be available for pairing. The region is then said to be less accessible. Thus, several authors have defined accessibility using the Gibbs free energy, which

reflects the strength of mRNA folding around the region of interest [14, 24, 28, 29, 31, 32]. Since, the MFE can be calculated through the computation of a partition function for base pairing, a more rigorous way to define accessibility is to use the partition function to compute the basepair probabilities [30, 76]. Despite the improvements over MFE, this partition function-based approach has not been widely adopted. (ii) Another way that accessibility has been defined is to use the partition function to compute the probabilities of bases being unpaired, or an equivalent pseudo energy called the opening energy [22]. In this work, we use this approach to define accessibility as it is mathematically well-defined and provides greater contextual information (See Figs 1 and 4A). Furthermore, an efficient algorithm to compute opening energy is implemented in the RNAplfold of the ViennaRNA suite [77]. A short comparison of these methods is given in Additional notes in S2 File.

Terai and Asai (2020) and ourselves have independently discovered that modelling the accessibility of translation initiation sites using the base-unpairing approach is superior to the simplistic MFE estimation [33, 78] (see Fig 2A for example). A key difference between our approaches is that we used RNAplfold which is based on a biophysical model of RNA with Turner's nearest-neighbor parameters [79], whereas Terai and Asai (2020) used Raccess, which is based on a probabilistic model and uses a further optimised set of Turner's parameters [80, 81].

Besides, very few applications of accessibility have been developed, for example, as implemented in RNAup and IntaRNA for the prediction of RNA-RNA intermolecular interactions [69, 77, 82]. We have advanced our findings by developing a web service and a command line tool for tuning protein expression called TIsigner. The underlying JavaScript and Python modules that we developed for the web service and command line tool, respectively, are open source and freely available (see Code and data availability).

Our method requires only a modest number of synonymous codon changes to the 5′ coding region of a given sequence (Fig 5). In contrast, other gene optimisers incorporate several features that require synonymous codon changes of almost the entire sequence.

## Implementations of TIsigner for improving recombinant protein production

Our TIsigner web service offers several unique features and supports recombinant protein expression in *E. coli*, *S. cerevisiae*, and *M. musculus* (optimisation regions −24:24, −7:89 and −−8:11, respectively; see Fig 2). Users can easily change the optimisation region to accommodate other expression hosts. Our TIsigner web service also allows full-length sequence optimisation in addition to the first nine codons. For *E. coli* hosts, users are warned when terminators or any custom sequence motifs are detected.

Furthermore, we provide a holistic solution to design experiments for recombinant protein production [83]. Importantly, TIsigner is integrated with SoDoPE and Razor web services that allow solubility optimisation and signal peptide prediction, respectively [84, 85]. Such integration allows a seamless transition between these three services. For example, a protein sequence of interest can be first submitted to Razor. If a signal peptide is detected, users have an option to check for solubility using SoDoPE and select the mature region using the interactive interface. Otherwise, users can also check the solubility of the full-length sequence. If protein domains are detected, users can consider optimising solubility by selecting any subregions. Regions with optimised solubility are instantly returned. Users can then redirect any selected region to TIsigner for accessibility optimisation. In contrast to the existing gene optimisers, their features are very limited [68, 86–88].

## Concluding remarks

The strengths of our approaches are five-fold. Firstly, the likelihood of success or failure can be assessed prior to running an experiment. Users can compare the opening energies calculated for the input and optimised sequences and the distributions of the 'success' and 'failure' of the PSI:Biology targets. We also introduced a scoring scheme in TIsigner to score the input and optimised sequences based upon how likely they are to be expressed (S5 Fig; see also Methods). Secondly, optimised sequences can have up to the first nine codons substituted (by default), meaning that gene optimisation can be done using PCR. For cloning, we propose a nested PCR approach, in which the final PCR reaction utilises a forward primer designed according to the optimised sequence [89] (Fig D in S7 Fig). Thirdly, the cost of gene optimisation can be reduced dramatically as gene synthesis is replaced with PCR using our approach. This enables high-throughput protein expression screening using the optimised sequences, generated at a low cost. Fourthly, tunable expression is possible, i.e. high, intermediate or even low expression 5′ codon sequences can be designed, allowing for more control over heterologous protein production, as demonstrated by our experiments (Fig 5). Finally, our fast, lightweight, stochastic simulation approach has opened up new avenues to study several aspects of gene expression, such as transcription, translation, cellular growth, and turnovers, which give good proxies to how cellular systems behave.

## Methods

### Plasmids

Plasmids were constructed using the MIDAS Golden Gate cloning system [90] (see Additional Methods, Figs A-E and Tables A-C in S2 File).

### Data

Datasets used in this study are listed in S1 File. These include fluorescence reporter expression datasets previously generated using *E. coli*, *Saccharomyces cerevisiae*, and *Mus musculus* cultured cells (S1 Fig), and recombinant protein production dataset from the Protein Structure Initiative: Biology (PSI:Biology; S2 Fig). Representative sequences were chosen from the *E. coli* green fluorescence protein (GFP) reporter dataset [16] using CD-HIT-EST [91, 92]. Two ribosome profiling libraries previously generated using *E. coli* were retrieved from the Sequence Read Archive (SRR7759806 and SRR7759807) [93].

### Sequence features analysis

CAI, tAI and Codon Context (CC) were calculated using the reference weights from Sharp and Li [11], Tuller et al. [50] and Ang et al. [51], respectively. Translation elongation rate was predicted using Iχnos [52] that was trained using an *E. coli* ribosome profiling dataset (S3 Fig). Local G+C content was also examined (S4 Fig).

   We use the minimum free energy (MFE), opening energy and avoidance as thermodynamic features of a mRNA. The MFE of a sequence is the energetically most optimal sequence in a Boltzmann's ensemble. Consequently this feature is based on a single structure. In contrast, the opening energy of a stretch *i..j* of nucleotides is the pseudo energy required to unpair that region considering all sub-optimal structures and is given by:

$$Opening\ energy = -kT\ ln\frac{Z_{unpaired}}{Z},\ \ \ \ \ \ \ (1)$$

where the where $k$ is the Boltzmann's constant and $T$ is the absolute temperature, $Z_{unpaired}$ is

the canonical partition function of the region $i..j$ and $Z$ is the total canonical partition function. The ratio $Z_{unpaired}/Z$ is the probability that the nucleotides $i..j$ are unpaired, note that this ratio incorporates base-unpairing probabilities for the larger region $(i - W)..(j + W)$ (Fig 1), where $W$ is the window size used to compute partition functions. See also the expanded equation in Supporting Information (S2 File).

A crucial difference between opening energy and the widely used MFE [14–16] is that $Z_{unpaired}$ captures all possible optimal and suboptimal structures in the Boltzmann's ensemble, where the nucleotides $i..j$ are unpaired, including possible pairings from the surrounding region ($W$). For a sufficiently large window ($W$), this approach captures the bulk of RNA contacts, even for very large RNAs [94]. In other words, all possible optimal or suboptimal structures beyond the target region are accounted for, resulting in a full ensemble average energy for a region (see a case study in the Results section, Accessibility captures the full ensemble average energy of a sequence). This is distinct from the traditional MFE approach that returns a single solution (optimal or near-optimal structure) for a given region.

The avoidance metric measures the number of potential intermolecular misinteractions, and is calculated by computing the possible interactions among RNAs. The conventional way to model interactions is to assume a two step process where nucleotides unpair and then hybridise. Thus the total interaction energy $\Delta G_{int}$ is given by:

$$\Delta G_{int} = \Delta G_u + \Delta G_h, \tag{2}$$

where $\Delta G_u$ is the opening energy and $\Delta G_h$ is the hybridisation energy. In the previous study, $\Delta G_u$ was inadvertently upweighted. In this study we correctly address this by using only the $\Delta G_h$ component, where $\Delta G_h < \Delta G_u$.

The computation of the partition function and related energetic quantities involve dynamic programming. The implementation of these dynamic programming algorithms is available on several tools such as ViennaRNA and IntaRNA [77, 82].

We used RNAfold, RNAplfold and RNAup from the ViennaRNA package (version 2.4.11) to calculate MFE, opening energy and avoidance, respectively [22, 69, 77, 95–98]. RNAfold was run with default parameters. Based on previous studies, we calculated MFE using the mRNA region -30:30 [16, 37–40]. For RNAplfold, sub-sequences were generated from the input sequences to calculate opening energies (using the parameters -W 210 -u 210), in practise there is a subtle difference between the u and W parameters, but for this work we set these to be equal (https://www.tbi.univie.ac.at/RNA/RNAplfold.1.html) [22]. For RNAup, we examined the stochastic interactions between the region 1:30 of each mRNA and 54 non-coding RNAs (using the parameters -b -o). RNAup reports the total interaction between two RNAs as the sum of energy required to open accessible sites in the interacting molecules $\Delta G_u$ and the energy gained by subsequent hybridisation $\Delta G_h$ [69]. For the interactions between each mRNA and 54 non-coding RNAs, we chose the most stable mRNA:ncRNA pair to report an inappropriate mRNA:ncRNA interaction, i.e. the pair with the strongest hybridisation energy, $(\Delta G_h)_{min}$.

## Simulation

To better understand the dynamics between accessibility and protein production, we performed a stochastic simulation using constructs with increasing opening energy on a simulated cellular system.

To set the simulation, we binned the opening energies between 2 and 32 kcal/mol in intervals of two, with each bin representing a 'reporter plasmid construct' whose opening energy is the mean of the bin. For each construct, 'technical replicates' were generated by allowing slight

variations on the mean opening energy of the bin. This is to model variation between replicates, and the discrepancies between the estimated and the actual opening energies in vivo. For each round of transcription, mRNA copies were randomly generated from 30 to 60 plasmid DNA copies [3, 99, 100]. Based upon our analysis of targets from PSI:Biology (S7 Fig), we chose an optimum opening energy of 12 kcal/mol or less for translation. However, this is probabilistic which occasionally allows protein production from higher opening energy transcripts. We allowed mRNA to decay probabilistically when a mRNA molecule is translated for more than 10 times.

To simulate protein toxicity, we set a threshold of protein to be 1,000,000 copies where the copy number of endogenous proteins is usually less than 10,000 [10]. Beyond this limit, a sporadic death of cells is simulated. However, in this simulation, the chance of staying viable and reproducing is higher than death, and cells grow steadily. This threshold also simulated random but low cell deaths in the experiment, without setting an extra variable.

To limit the computational complexity of the simulation, we use smaller constants and iterations. Initialising with 100 cells, the algorithm was set to terminate either after 10,000 iterations or when the total number of cells is zero. After termination, the total number of proteins and cells for each construct were taken from the endpoints. To imitate 'biological replicates', we repeated the above simulation three times with different random numbers, which provides slightly different initial conditions for each experiment.

## Development of Translation Initiation coding region designer (TIsigner)

Finding a synonymous sequence with a maximum accessibility is a combinatorial problem that spans a vast search space. For example, for a protein-coding sequence of nine codons, assuming an average of 3 synonymous codons per amino acid, we can expect a total of 19,682 unique synonymous coding sequences. This number increases rapidly with increasing numbers of codons. Heuristic optimisation approaches are preferred in such situations because the search space can be explored more efficiently to obtain nearly optimal solutions.

To optimise the accessibility of a given sequence, TIsigner uses a simulated annealing algorithm [101–104], a heuristic optimisation technique based on the thermodynamics of a system settling into a low energy state after cooling. Simulated annealing algorithms have been used to solve many combinatorial optimisation problems in bioinformatics. For example, we previously applied this algorithm to align and predict non-coding RNAs from multiple sequences [105]. Other studies use this algorithm to find consensus sequences [103], optimise ribosome binding sites [24] and predict mRNA foldings [106] using MFE models.

According to statistical mechanics, the probability $p_i$ of a system occupying energy state $E_i$, with temperature $T$, follows a Boltzmann distribution of the form $e^{E_i/T}$, which gives a set of probability mass functions along every point $i$ in the solution space. Using a Markov chain sampling, these probabilities are sampled such that each point has a lower temperature than the previous one. As the system is cooled from high to low temperatures ($T \rightarrow 0$), the samples converge to a minimum of $E$, which in many cases will be the global minimum [103]. A frequently used Markov chain sampling technique is Metropolis-Hastings algorithm in which a 'bad' move $E_2$ from initial state $E_1$ such that $E_2 > E_1$, is accepted if $R(0, 1) \geq p_2/p_1$, where $R(0, 1)$ is a uniformly random number between 0 and 1.

In our implementation, each iteration consists of a move that may involve multiple synonymous codon substitutions. The algorithm begins at a high temperature where the first move is drastic, synonymous substitutions occur in all replaceable codons. At the end of the first iteration, a new sequence is accepted if the opening energy is smaller than that of the input sequence. However, if the opening energy of a new sequence is greater than that of the input

sequence, acceptance depends on the Metropolis-Hastings criteria. The accepted sequence is used for the next iteration, which repeats the above process. As the temperature cools (exponentially decreasing), the moves get milder with fewer synonymous codon changes (S7 Fig). Simulated annealing stops upon reaching a near-optimum solution.

For the web version of TIsigner, the default number of replaceable codons is restricted to the first nine codons. However, this default setting can be reset to range from the first four to nine codons, or the full length of the coding sequence. Since the accessibility of a fixed region is optimised, this process only takes $\mathcal{O}(1)$ time (S10 Fig). Furthermore, TIsigner runs multiple simulated annealing instances, in parallel, to obtain multiple possible sequence solutions.

When users select T7lac promoter as the 5′UTR, they can adjust 'Expression Score', that is calculated based on the PSI:Biology dataset (see below). This allows them to tune the expression level of a target gene. In contrast, when users input a custom 5′UTR sequence, they only have the option to either maximise or minimise expression.

To implement 'Expression Score', the posterior probabilities of success for input and optimised sequences are evaluated using the following equations from Bayesian statistics:

$$positive\ posterior\ odds = prior\ odds \times fitted\ positive\ likelihood\ ratio, \tag{3}$$

$$positive\ posterior\ probability = \frac{positive\ posterior\ odds}{1 + positive posterior odds}, \tag{4}$$

The fitted positive likelihood ratios in Eq (3) were obtained from the following 4-parametric logistic regression equation:

$$fitted\ positive\ likelihood\ ratio = d + \frac{a - d}{1 + \left(\frac{positive\ likelihood\ ratio}{c}\right)^{b}}, \tag{5}$$

with parameters a, b, c, and d. The prior probability was set to 0.49, which is the proportion of 'Expressed' (N = 21,046) divided by 'Cloned' (N = 42,774) of the PSI:Biology targets reported as of 28 June 2017 (http://targetdb.rcsb.org/metrics/). Posterior probabilities were scaled as percentages to score the input and optimised sequences (S5 Fig).

The presence of terminator-like elements [107] in the protein-coding region may result in expression of truncated mRNAs due to early transcription termination. Therefore, we implemented an optional check for putative terminators in the input and optimised sequences by cmsearch (INFERNAL version 1.1.2) [108] using the covariance models of terminators from RMfam [109, 110]. We also allow users to filter the output sequences for the presence of restriction sites. Restriction modification sites (AarI, BsaI, and BsmBI) are avoided by default.

Besides *E. coli*, users can choose *S. cerevisiae*, *M. musculus* or 'Other' as the expression host. The regions for optimising accessibility are −7:89, −8:11 and −24:89 for *S. cerevisiae*, *M. musculus* and 'Other', respectively (Fig 2 and S1 Fig). When users choose 'Custom' for expression host, the region for optimising accessibility becomes customisable.

## Sequence optimisation

To compare accessibility between sequences optimised using TIsigner and other tools, we submitted the PSI:Biology targets that failed to be expressed (N = 2,650) to the ExpOptimizer web server from NovoPro Bioscience (https://www.novoprolabs.com/tools/codon-optimization). A total of 2,573 sequences were optimised. The target sequences were also optimised using a local version of COOL [26] and TIsigner using default settings. We also ran a random synonymous codon substitution as a control for these 2,573 sequences.

## GFP assay

BL21(DE3)pLysS competent *E. coli* cells (Invitrogen) were transformed with plasmids and grown overnight on Luria-Bertani (LB) agar plates containing spectinomycin (50 $\mu$g/ml) and chloramphenicol (25 $\mu$g/ml). Single colonies were picked and inoculated into 3 ml LB broth containing the same antibiotics, and cultures were grown for 18 hours at 37˚C, 200 rpm. Cultures were diluted with fresh media at 1:20 and grown at 37˚C, 200 rpm, until reaching the mid-logarithmic growth phase (optical densities at 600 nm (OD600) of 0.3). Of each culture, 20 $\mu$l was seeded into 96-well plates containing 180 $\mu$l LB broth supplemented with antibiotics and isopropyl-$\beta$-D thiogalactopyranoside (IPTG) (1 mM final concentration) per well. Fluorescence intensities and ODs were measured in a black, flat, clear bottom 96-well plate with lid (CELLSTAR, Greiner) using a FLUOstar Omega plate reader (BMG Labtech) equipped with an excitation filter (band pass 485–12) and an emission filter (band pass 520) for GFP and excitation filter (band pass 484) and an emission filter (band pass 610–10) for mScarlet-I. The plate was incubated at 37˚C with "meander corner well shaking" at 300 rpm for 7 hours measuring fluorescence and ODs every 10 minutes. Fluorescence was measured in a 2 mm circle recording the average of 8 measurements per well. Average values of technical replicates were calculated and normalised to the mScarlet-I second reporter, and then to the normalised value of the GFP variant with the highest opening energy (21.68 kcal/mol). Normalised fluorescence values were obtained from the average values of biological replicates (S8 Fig and S3 File).

## Luciferase assay

BL21Star(DE3) competent cells (Invitrogen) were transformed with plasmids and grown overnight at 37˚C on LB agar plates containing 50 $\mu$g/ml spectinomycin. Single colonies were picked and inoculated into 5 ml LB broth (50 $\mu$g/ml spectinomycin) and grown for 18 hours at 37˚C, 200 rpm. Bacterial cultures were diluted with fresh media at 1:20 and grown at 37˚C, 200 rpm, up to a mid-logarithmic phase (OD600 of 0.4). The cultures were split and induced with IPTG at a final concentration of 0.25 mM (or uninduced as controls), and seeded into a white, flat, clear bottom 96-well white plate with lid (Costar, Corning), 150 $\mu$l per well, in triplicates. Cells were incubated in a FLUOstar Omega Microplate Reader (BMG LABTECH) for 90 minutes at 25˚C, 200 rpm, and OD600 was measured every 15 minutes (over 7 cycles). Cells were harvested by centrifugation at 3000 ×g, for 10 minutes, at 20˚C. Supernatants were removed. As the substrate can penetrate into cells, 50 $\mu$l of coelenterazine h (Promega) was added to living cells to minimise sample processing steps and variability [111, 112]. Luminescence was measured ($\lambda_{em}$ = 475 nm) in a Clariostar microplate reader (BMG LABTECH) at 25˚C every 2 minutes (over 11 cycles). Average values of technical replicates were calculated and normalised to the wild-type. Normalised luminescence values were obtained from the average values of biological replicates (S9 Fig and S3 File).

## Statistical analysis

AUC and Gini importance scores were calculated using scikit-learn (version 0.20.2) [113]. The 95% confidence intervals for AUC scores were calculated using DeLong's method [114]. Spearman's correlation coefficients and Kolmogorov-Smirnov statistics were calculated using Pandas (version 0.23.4) [115] and scipy (version 1.2.1) [116, 117], respectively. Positive likelihood ratios with 95% confidence intervals were calculated using the bootLR package [118, 119]. The P-values of multiple testing were adjusted using Bonferroni's correction and reported to machine precision. Plots were generated using Matplotlib (version 3.0.2) [120] and Seaborn (version 0.9.0) [121].

## Supporting information

**S1 Fig. Heatmaps of correlations between opening energies and protein abundances for each of the sub-sequence regions (related to Fig 2).** Green unfilled triangles indicate the regions before and after scaling (left and right panels, respectively). A: For *E. coli*, we used a representative GFP expression dataset from Cambray et al. (2018) [16]. The reporter library consists of GFP fused in-frame with a library of 96-nt upstream sequences (N = 14,425). B: For *S. cerevisiae*, we used a YFP expression dataset from Dvir et al. (2013) [19]. The YFP reporter library consists of 2,041 random decameric nucleotides inserted at the upstream of YFP start codon. C: For *M. musculus*, we used the GFP expression dataset from Noderer et al. (2014) [34]. The GFP reporter library consists of 65,536 random hexameric and dimeric nucleotides inserted at the upstream and downstream of GFP start codon, respectively. $R_s$, Spearman's rho. (PDF)

**S2 Fig. Expression outcomes of the PSI:Biology targets in *E. coli* (related to Figs 3C and 4).** A total of 11,430 PSI:Biology targets from over 189 species were analysed in this study (N = 8,780 and 2,650, 'success' and 'failure' groups, respectively). Genera with at least 20 target genes are shown and the remaining as 'Others'. The top three PSI:Biology targets are from four *Pseudomonas*, five *Bacillus* and six *Clostridium* species. Red asterisk, obelisk and diesis indicate *Homo sapiens*, *S. cerevisiae* and *E. coli*, respectively. These target genes were inserted into the pET21_NESG expression vector, in which the promoter and fusion tag are T7lac and C-terminal His tag, respectively. (PDF)

**S3 Fig. Ribosome footprints in 25-nt fragments show a strong triplet periodicity, indicating translation (related to Fig 4).** These 25-nt footprints (green unfilled rectangle) were used to train a neural network model [52] in order to predict the translation elongation rates of the PSI:Biology targets. Ribosome profiling data (SRR7759806 and SRR7759807 [93]) were first aligned to *S. cerevisiae* transcriptome. SAM alignment files were merged, and ribosome footprints which were mapped to each frame were enumerated. See https://github.com/Gardner-BinfLab/TIsigner_paper_2019. FP, footprints. (PDF)

**S4 Fig. Analysis of the local G+C contents in the PSI:Biology target genes (related to Fig 4).** A: The G+C contents in the regions −24:24 and −30:30 weakly correlate with opening energy and MFE, respectively. Green unfilled squares indicate Spearman's correlations ($R_s$) between the local G+C contents and the corresponding local features. B: The local G+C contents show a similar prediction accuracy (AUC scores shown in parentheses). AUC, Area Under the receiver operating characteristic Curve; MFE, Minimum Free Energy. (PDF)

**S5 Fig. Opening energy of 10 kcal/mol or below at the region −24:24 is about two times more likely to come from the target genes that are successfully expressed than those that failed.** Cumulative frequency distributions of the true positive and false positive (less than type), and true negative and false negative (more than type) derived from the ROC analysis in Fig 4C (left panel, opening energy −24:24). These values were used to estimate positive likelihood ratios with 95% confidence intervals using 10,000 bootstrap replicates. The estimated ratios and/or confidence intervals are inaccurate at low numbers of true positives or true negatives. Therefore, a four-parameter logistic curve was fitted to the positive likelihood ratios. Fitted values are useful to estimate the posterior probability of protein expression. (PDF)

**S6 Fig. Accessibility is able to capture the full ensemble average energy of a sequence.** The levels of major capsid protein expressed by the wild-type (orange) and mutant (blue) strains of bacteriophage T7 [53]. The mutant major capsid gene was codon-deoptimised such that the first and the last 14 codons remained unchanged. Proteomic analyses were carried out at 1, 5 and 9 min post-infection in four biological replicates. Opening energy −24:24, MFE −30:30, and CAI of the wild-type and mutant sequences were compared. The approximated 'Expression Scores' of the wild-type and mutant sequences are 89 and 38, respectively (opening energies of 9.05 kcal/mol and 13.76 kcal/mol, respectively). MFE, Minimum Free Energy; CAI, Codon Adaptation Index.
(PDF)

**S7 Fig. Accessibility of translation initiation sites can be increased by synonymous codon substitution within the first nine codons using simulated annealing.** A: Schedules in simulated annealing. The ratio of temperature to the number of the first N substitutable codons decreases exponentially with increasing number of iterations. B: Accessibility of translation initiation sites increases with increasing number of the first N replaceable codons. The PSI: Biology targets that failed to be expressed were optimised using simulated annealing (N = 2,650). The Kolmogorov-Smirnov distance between the distributions of '9' and 'full-length' was significantly different but sufficiently close ($D_{KS} = 0.09$, $P < 10^{-7}$), indicating that optimisation of the first nine codons can achieve nearly optimum accessibility. For comparison, the distribution of the PSI:Biology targets that were successfully expressed are shown (N = 8,780). See also S1 File. C: Accessibility of translation initiation sites can be increased indirectly using the existing gene optimisation tools and random synonymous codon substitution. 'TIsigner (9)' refers to the default settings of our tool, which allows synonymous substitutions up to the first nine codons (as above). D: Accessibility of translation initiation sites can be optimised using PCR. The forward primer should be designed according to TIsiger optimised sequences. For example, using a nested PCR approach, the optimised sequence can be produced using the forward primer designed with appropriate mismatches (gold bulges) to amplify the amplicon from the initial PCR reaction.
(PDF)

**S8 Fig. Luciferase reporter assay (related to Fig 5A).** A: SDS-PAGE gel shows the protein bands of *Renilla* luciferase (RLuc) in the soluble and insoluble fractions of BL21Star(DE3) lysates. The expression of RLuc can be improved, despite its poor solubility in *E. coli*. Selected bacterial clones were grown at 25°C, 200 RPM. The solubilities of wildtype (WT) RLuc and designed variants were compared after 4-hour IPTG induction. The blue and red arrows (about 36kDa) indicate that RLuc was poorly soluble. No RLuc protein bands were detected from the uninduced cultures and IPTG-induced negative control (empty vector control that lacks Rluc gene and T7lac promoter). B: The luciferase activities of commercially designed RLuc reporter genes (full-length sequence optimisation) and a TIsigner optimised sequence (9.9 kcal/mol) are significantly higher than the wild-type luciferase (Mann-Whitney U tests, $P = 9.1 \times 10^{-3}$). Opening energies are shown next to labels. IPTG, isopropyl-β-D thiogalactopyranoside.
(PDF)

**S9 Fig. The yields of an antibody fragment and an archaebacterial dioxygenase can be improved by synonymous codon changes within the first six codons (related to Fig 5A).** A RTS *E. coli* cell-free expression system was previously used to express these recombinant proteins [30]. The expression levels are shown in arbitrary units (AU) based on the densitometric analysis of previously published Western blots (S3 File). WT, wild-type.
(PDF)

**S10 Fig. Sequence length does not affect software performance because only a fixed region is taken into account during optimisation ($\mathcal{O}(1)$ time).**
(PDF)

**S1 File. Datasets used in this study and results.** Results for Fig 4D, and Figs B and C in S7 Fig.
(XLSX)

**S2 File. Additional notes and methods.** Figs A-E, and Tables A-C.
(PDF)

**S3 File. Experimental results.** GFP and RLuc reporter assay results and densitometric data for Fig 1A, Fig B in S8 and S9 Figs.
(XLSX)

## Acknowledgments

We thank Professor Ivo Hofacker for fruitful discussions at the Benasque RNA Meeting, and Dr Ronny Lorenz for helpful discussions about RNAplfold. We are grateful to the members of the Biomolecular Interaction Centre at the University of Canterbury for supporting this research. We thank New Zealand eScience Infrastructure for providing high performance computing resources.

## Author Contributions

**Conceptualization:** Bikash K. Bhandari, Chun Shen Lim, Craig van Dolleweerd, Paul P. Gardner.

**Data curation:** Chun Shen Lim.

**Formal analysis:** Bikash K. Bhandari, Chun Shen Lim, Daniela M. Remus, Augustine Chen, Craig van Dolleweerd.

**Funding acquisition:** Craig van Dolleweerd, Paul P. Gardner.

**Investigation:** Bikash K. Bhandari, Chun Shen Lim, Daniela M. Remus, Augustine Chen, Craig van Dolleweerd, Paul P. Gardner.

**Methodology:** Bikash K. Bhandari, Chun Shen Lim, Daniela M. Remus, Augustine Chen, Craig van Dolleweerd, Paul P. Gardner.

**Project administration:** Chun Shen Lim, Daniela M. Remus, Augustine Chen, Craig van Dolleweerd, Paul P. Gardner.

**Resources:** Craig van Dolleweerd, Paul P. Gardner.

**Software:** Bikash K. Bhandari, Chun Shen Lim.

**Supervision:** Chun Shen Lim, Craig van Dolleweerd, Paul P. Gardner.

**Validation:** Bikash K. Bhandari, Chun Shen Lim, Daniela M. Remus, Augustine Chen.

**Visualization:** Bikash K. Bhandari, Chun Shen Lim, Daniela M. Remus, Augustine Chen.

**Writing – original draft:** Bikash K. Bhandari, Chun Shen Lim, Daniela M. Remus, Augustine Chen, Craig van Dolleweerd, Paul P. Gardner.

**Writing – review & editing:** Bikash K. Bhandari, Chun Shen Lim, Daniela M. Remus, Augustine Chen, Craig van Dolleweerd, Paul P. Gardner.

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
