## [Decision Letter · Decision Letter 0]

5 Aug 2021

Dear Dr Gardner,

Thank you very much for submitting your manuscript "Analysis of 11,430 recombinant protein production experiments reveals that protein yield is tunable by synonymous codon changes of translation initiation sites" for consideration at PLOS Computational Biology.

As with all papers reviewed by the journal, your manuscript was reviewed by members of the editorial board and by several independent reviewers. In light of the reviews (below this email), we would like to invite the resubmission of a significantly-revised version that takes into account the reviewers' comments.

The authors addressed several critiques outlined in initial reviews. However Ref 2 raises issuea that are fundamntal to claims of novelty and significance presented in this paper. The authors should have another chance to address them with subsequent re-review by Ref.2.

We cannot make any decision about publication until we have seen the revised manuscript and your response to the reviewers' comments. Your revised manuscript is also likely to be sent to reviewers for further evaluation.

Sincerely,

Eugene I. Shakhnovich

Guest Editor

PLOS Computational Biology

Nir Ben-Tal

Deputy Editor

PLOS Computational Biology

The authors addressed several critiques outlined in initial reviews. However Ref 2 raises issuea that are fundamntal to claims of novelty and significance presented in this paper. The authors should have another chance to address them with subsequent re-review by Ref.2.

Reviewer's Responses to Questions

**Comments to the Authors:**

Reviewer #1: The authors have adequately responded to my review. I have no further concerns.

Reviewer #2: The clarifications provided by the authors, in particular the definition of the unfolding energy, are helpful. However, the changes do not satisfactorily address the concerns laid out in the original review. First, the relationship between the proposed method and existing approaches in the literature is inaccurately described, and the improvement over existing approaches is thus insufficiently demonstrated (original point 2). Second, the mechanistic interpretations are not sufficiently justified (original points 3 and 4). These concerns are described further below.

Concerning point 2, there appears to be confusion regarding the relationship between free energies and partition functions. What the authors call opening energy is the free energy difference between two ensembles of structures, one with partition function Z_unpaired and the other with partition function Z. To clarify, there are two types of free energies that appear in RNA structure calculations. The first is the free energy of a particular structure labeled i, g_i. The MFE is equal to the minimum g_i of the all possible structures considered by the RNA folding model. The second free energy is related to the partition function of an ensemble of structures: G = -kT log Z, where Z = sum_i exp(-g_i/kT). Free energy differences, delta G = G_2 - G_1, between two ensembles of structures with partition functions Z_1 and Z_2 are physically meaningful.

The delta G used in the current ref 64 is also a free energy difference defined between two ensembles of structures, although the definition of these ensembles is slightly different from the unfolding energy used here. Thus, these methods are very closely related. Because of the direct relationship between free energies of structure ensembles and the corresponding partition functions, there is no qualitative distinction between the approach used in this work and those used in some previous studies. The revised text "This partition function based approach is distinct..." is consequently misleading. Furthermore, the idea that calculations using ensembles of RNA structures can predict the probability of specific bases being unpaired is also not new. For example, ref 64 also makes use of the probability that a particular base is unpaired, which is equal to Z_{i unpaired}/Z. In my view, the paper will be strengthened by accurately describing how the proposed method differs from previously published approaches.

Despite the fact that using ensemble calculations and unpairing probabilities is not new, it is quite possible that the precise definitions of the ensembles used by the authors provide better predictions of protein abundance than existing methods. To the best of my knowledge, the particular free energy difference that the authors use here has not been examined elsewhere. If this particular definition is superior to those examined previously, then this is a useful contribution. However, without making a direct comparison between the proposed quantity and the existing ensemble-based methods in the literature, the authors do not make a convincing case for adopting their proposed method.

Concerning point 3, the authors' speculation as to why the highest correlation is achieved when unpairing ~50 bases seems plausible. However, the analysis here does not provide specific evidence in direct support of the conclusion that accessibility for translation initiation is the central mechanism at play. For example, convincing evidence might involve showing that the Shine--Dalgarno sequence (in E. coli) or the start codon are specifically implicated by the analysis. Thus, in my view, existing studies in the literature provide much stronger support for the translation-initiation mechanism.

Concerning point 4, free energy differences calculated by RNA structure prediction methods do indeed have physical meanings. As clarified above, free energy differences (not pseudo energies) are related to partition function ratios by Delta G = -kT log (Z_2 / Z_1). Therefore, the interpretation of the free energy difference is that exp(-Delta G / kT) is the relative probability of structure ensembles 1 and 2. Of course, the RNA folding models have been parameterized for solution conditions (e.g., ionic strength and temperature) that may not match physiological conditions, the models make assumptions that restrict the allowed RNA structures, and the influence of the complex intracellular environment is completely neglected; thus, it is likely that the calculated free energy differences are approximate. Using these numbers as proxies for weak versus strong base pairing, as the authors suggest, is thus completely fine.

Nonetheless, the interpretation as the relative probability of structure ensembles becomes important when arguing in favor of a mechanism, such as the translation-initiation mechanism. Specifically, the opening energy is related to the probability that a subset of bases are unpaired. When considering a region of ~100 bases and a free energy difference in excess of 20 kcal/mol (e.g., Figure 1B), one has to ask whether it is physically reasonable for this quantity to be related to the proposed translation initiation mechanism. If the unfolding energy is being proposed solely as a proxy for designing optimal sequences, as the authors now state, then there is no issue with the interpretation of the actual value. However, if the authors want to use these calculations to support an argument in favor of a specific mechanism, as the authors claim elsewhere in the paper, then it is necessary to reconcile the magnitudes of these free energy differences with the proposed mechanism as well as to explain why the magnitudes differ so much from one organism to the next.

Reviewer #3: The authors have addressed several comments, most importantly now provided a detailed procedure of calculating opening energies. Unfortunately, after going through the definition, it seems that ‘opening energy’ is very similar to the DGunfold parameter, or more importantly the ‘punbound’ parameter used in reference 68 (now ref 64). This paper also arrived at the same conclusion that accessibility of the SD region is important to dictate mRNA and protein expression. As I mentioned previously, it is well established from several previous work that keeping the 5’-terminus of the mRNA sequence free of structure is important to allow efficient ribosome binding and translation initiation, and this is precisely their finding. Hence, the question about the novelty of the work remains. Additionally, in response to Reviewer 1 the authors say that ref 68 does not use a partition function-based methodology, but that is not correct.

As Reviewer 1 also mentions, the authors should calculate MFE by varying the length of the subsequences, otherwise the comparison between MFE and opening energy remains invalid. I didn’t understand what the authors meant while saying that ‘MFE is usually calculated for a certain region and is a single value’.

About my question as to why the region of the mRNA used for calculation is so different in different organisms, the authors say that it is because the SD region is absent in other eukaryotes. That can change the length of the sequence upstream of AUG, but why later?

Despite this, I think TIsigner is a useful program to optimize expression by modification of synonymous codons, but I don’t really see much beyond that.

**Have the authors made all data and (if applicable) computational code underlying the findings in their manuscript fully available?**

Reviewer #1: Yes

Reviewer #2: Yes

Reviewer #3: Yes

PLOS authors have the option to publish the peer review history of their article (what does this mean?). If published, this will include your full peer review and any attached files.

Reviewer #1: No

Reviewer #2: No

Reviewer #3: No
---

## [Decision Letter · Decision Letter 1]

19 Sep 2021

Dear Dr Gardner,

We are pleased to inform you that your manuscript 'Analysis of 11,430 recombinant protein production experiments reveals that protein yield is tunable by synonymous codon changes of translation initiation sites' has been provisionally accepted for publication in PLOS Computational Biology.

Best regards,

Eugene I. Shakhnovich

Guest Editor

PLOS Computational Biology

Nir Ben-Tal

Deputy Editor

PLOS Computational Biology

Reviewer's Responses to Questions

**Comments to the Authors:**

Reviewer #2: The authors have sufficiently addressed my previous comments. The revised manuscript clarifies the computational methods and the conclusions that can be drawn from the correlation analyses. The discussion also adequately contextualizes the results with respect to previously published works.

Reviewer #3: Along with comments to my questions as well as Reviewer 2, I think that all queries have been answered. I have no more questions.

**Have the authors made all data and (if applicable) computational code underlying the findings in their manuscript fully available?**

Reviewer #2: None

Reviewer #3: None

PLOS authors have the option to publish the peer review history of their article (what does this mean?). If published, this will include your full peer review and any attached files.

Reviewer #2: No

Reviewer #3: No

---

## [Editor Report · Acceptance letter]

30 Sep 2021

PCOMPBIOL-D-21-01024R1 

Analysis of 11,430 recombinant protein production experiments reveals that protein yield is tunable by synonymous codon changes of translation initiation sites

Dear Dr Gardner,

I am pleased to inform you that your manuscript has been formally accepted for publication in PLOS Computational Biology. Your manuscript is now with our production department and you will be notified of the publication date in due course.

With kind regards,

Andrea Szabo
